# GNeRP: Gaussian-guided Neural Reconstruction of Reflective Objects with Noisy Polarization Priors

**Yang LI[1,2,4]**  **Ruizheng WU[4]**  **Jiyong LI[3]**  **Yingcong CHEN[1,2,†]**

[1]AI Thrust, HKUST(GZ), Nansha, Guangzhou, China
[2]Department of Computer Science & Engineering, HKUST, Clear Water Bay, HongKong SAR, China
[3]Department of Computer Science, Sun Yat-sen University, Panyu, Guangzhou, China
[4]R & D Center, SmartMore, Qianhai, Shenzhen, China
[†] Corresponding Author
yli803@connect.ust.hk, rzwu@cse.cuhk.hk, lijy373@mail2.sysu.edu.cn,
yingcongchen@ust.hk

## Abstract

Learning surfaces from neural radiance field (NeRF) became a rising topic in Multi-View Stereo (MVS). Recent Signed Distance Function (SDF)–based methods demonstrated their ability to reconstruct accurate 3D shapes of Lambertian scenes. However, their results on reflective scenes are unsatisfactory due to the entanglement of specular radiance and complicated geometry. To address the challenges, we propose a Gaussian-based representation of normals in SDF fields. Supervised by polarization priors, this representation guides the learning of geometry behind the specular reflection and captures more details than existing methods. Moreover, we propose a reweighting strategy in the optimization process to alleviate the noise issue of polarization priors. To validate the effectiveness of our design, we capture polarimetric information, and ground truth meshes in additional reflective scenes with various geometry. We also evaluated our framework on the PANDORA dataset. Comparisons prove our method outperforms existing neural 3D reconstruction methods in reflective scenes by a large margin.

## 1 Introduction

Reconstructing 3D shapes from 2D images (Furukawa et al., 2015) is a fundamental problem in computer vision and graphics, with downstream applications such as 3D printing (Chen & Yang, 2014), autonomous driving (Chen et al., 2017), and Computer Aided Design (Furukawa et al., 2010). Although diffuse objects are precisely reconstructed, reflective and textured-less scenes remain challenging. Traditional Multi-View Stereo (MVS) methods (Bregler et al., 2000) rely on stereo matching across views, which is hindered in the presence of specular surfaces and texture absence. Recent methods utilizing implicit neural representation learning for 3D reconstruction have shown promising accuracy (Mescheder et al., 2019; Yariv et al., 2021), yet they overlook the specular reflection between light rays and surfaces, failing to adequately handle glossy objects with high-frequency specular reflection.

Existing methods (Zhang et al., 2021; Liu et al., 2023; Dave et al., 2022) attempt to separate specular reflection components from radiance to improve the reconstruction process. These methods model the interaction of light rays and surfaces by Bidirectional Reflectance Distribution Functions (BRDFs) and estimate them by neural networks. However, the inverse problem posed by BRDFs formulation is highly ill-posed (Guo et al., 2014), and low-frequency bias (Xu et al., 2019) of neural BRDFs making the learned geometry over-smoothed (Liu et al., 2023). Therefore, high-frequency geometry with specular reflection shown in Fig. 1 (a) is intractable for them. Besides, a few methods employ polarization priors to facilitate the learning of specular reflection because they reveal information about surface normals. However, polarization information is always concentrated in

specular-dominant regions and noisy in diffuse regions (Kajiyama et al., 2023), making the reconstruction process in diffuse-dominant regions distorted.

Faced with the bias of neural BRDF and noise issues of polarization priors, we present a novel perspective for reconstructing the detailed geometry of reflective objects. Our key idea is to extend the geometry representation from scalar SDFs to Gaussian fields of normals supervised by polarization priors. Given a surface point, the normals within its neighborhood are approximated by a 3D Gaussian. And it's a more informative representation of geometry. The mean shows the overall (low-frequency) orientation of the surface, while the covariance captures high-frequency details. Coincidentally, the representation can be splatted into the image plane as 2D Gaussians, as illustrated in Fig. 1 (b). The splatting skips the disentangled specular radiance. Learning of the 2D Gaussians can be directly supervised by the polarization information about surface normals. Hence, it circumvents the separation of complex geometry and specular reflection and manages to learn detailed geometry.

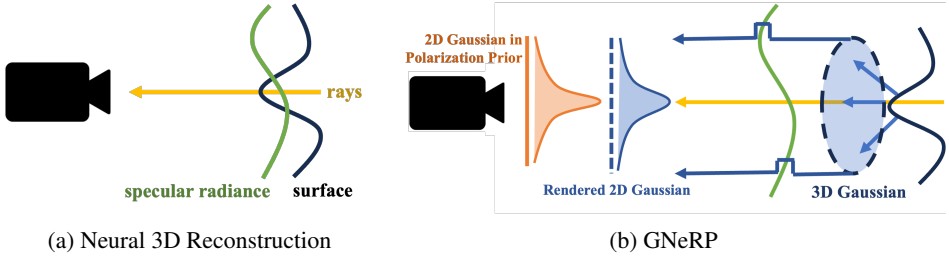

(a) Neural 3D Reconstruction          (b) GNeRP

Figure 1: Visualization of Gaussians of normals in Neural Reconstruction pipelines. 2D Gaussians can be rendered from 3D Gaussians of learned normals.

Furthermore, to tackle the noise issues of polarization priors, we introduce a Degree of Polarization (DoP) based reweighting strategy. This strategy adaptively balances the supervision of radiance and polarization priors, enhancing the reconstructing accuracy in diffuse-dominant regions.

In summary, our contributions are as follows:

- We propose a novel polarization-based Gaussian representation of detailed geometry to guide the learning of geometry behind specular reflection.

- We propose a DoP reweighing strategy to alleviate noise and imbalance distribution problems of polarization priors.

- We collect a new challenging multi-view dataset consisting of both radiance and polarimetric images with more diverse and challenging scenes.

## 2   RELATED WORK

### 2.1   MULTI-VIEW 3D RECONSTRUCTION

Traditional Multi-view Stereo focuses on the extraction of cross-view features to generate 3D points. (Schönberger et al., 2016; Galliani et al., 2015) try to estimate the depth map of the observed scene with multi-view consistency and fuse the depth maps into dense point clouds. These methods suffer from accumulating errors due to complex pipelines, and features are hard to be extracted from reflective objects. (Mescheder et al., 2019) explicitly models the objects' occupancy in a voxel grid to guarantee a complete object model is created. However, the resolution of the voxel limits the accuracy of the reconstructed surface. Recently, the success of NeRF (Mildenhall et al., 2020), which uses a simple MLP to encode the color and density information for a scene, inspired researchers to resort to implicit representation for multi-view 3D reconstruction. The representative works are Unisurf (Oechsle et al., 2021), NeuS (Wang et al., 2021), and VolSDF (Yariv et al., 2021), which exploit an MLP to model a Signed Distance Function (SDF) for a target scene. These methods optimize the implicit representation, i.e., SDF, by minimizing the MSE loss between the rendered pixel's radiance value and the corresponding pixel's radiance value in GT images. Such a paradigm works

well with Lambertian surfaces. However, only view-conditioned radiance fields fail in reflective scenes.

## 2.2 BRDF FOR REFLECTIVE OBJECTS RECONSTRUCTION

In the regions with complex geometry, BRDFs always exhibit high-frequency variations due to the normals terms, while the low-frequency implicit bias of neural networks (Xu et al., 2019) disables neural BRDFs from predicting these abrupt changes. It always results in over-smoothed geometry. For example, NeRO (Liu et al., 2023) adopts Micro-facet BRDF (Cook & Torrance, 1981) parameterized by material and normal distribution terms. Although its results of smooth mirror-like objects are excellent, the spatial continuity of neural BRDF is a barrier to the combination of complex geometry and specular reflection. In the regions with complex geometry, sole multi-view images with disentangled radiance result in severe ill-posedness of the inverse problem, as is shown in Fig. 1 (a). Moreover, explicit estimation of anisotropic normals distribution has been used in rendering delicate objects, such as anisotropy shading of hairs (Banks, 1994), to improve the perception of orientation and shapes (Ament & Dachsbacher, 2015). However, anisotropic normals distribution in neural SDFs for 3D reconstruction remains under-defined and non-trivial. Our method proposes 3D Gaussians, of which anisotropic 3D covariance is more informative than the scalar normals distribution term in NeRO. The latter only measures the concentration of normals at a surface point.

## 2.3 MULTI-VIEW 3D RECONSTRUCTION WITH POLARIZATION

Polarization prior reveals the azimuth angle of the surface normal, i.e., the angle between the normal projection onto the image plane and the positive x-axis of the image. Shape-from-polarization has been investigated by other papers (Atkinson & Hancock, 2006; Foster et al., 2018; Fukao et al., 2021; Cui et al., 2017; Kadambi et al., 2015; Zhao et al., 2020) before the invention of neural 3D reconstruction. But most of them are focused on common scenes. For example, PMVIR (Zhao et al., 2020) exploits the relation of the polarization angle and the azimuth angle of normals but with only Lambert shading, and thus it cannot treat reflective objects at all. Neural 3D Reconstruction with polarization priors has also been explored. Sparse Ellipsometry (Hwang et al., 2022) develops a device to capture polarimetric information and 3D shapes concurrently. However, their reconstruction is always disturbed by the noise in diffuse-dominant regions. For example, PANDORA (Dave et al., 2022) extends radiance in BRDF into polarimetric dimensions while the geometry of diffuse regions cannot be learned properly.

## 2.4 GAUSSIANS IN 3D SCENE REPRESENTATION

Gaussians are used to represent the attributes of 3D scenes. Mip-NeRF (Barron et al., 2021) encodes Gaussian regions of space rather than infinitesimal points for anti-aliasing. (Zwicker et al., 2001) proposes Gaussian splatting that taking volume data as 3D Gaussians and nearly projects the 3D Gaussian to the 2D one (Kerbl et al., 2023). (Kerbl et al., 2023) implements the splatting pipeline on the NeRF for real-time rendering. In numerical geometry, (Berkmann & Caelli, 1994) calculates the covariance matrix from the projections of the normal vectors to highlight the edges and local geometry of surfaces. Inspired by them, we demonstrate a further fact that taking surface normals as 3D Gaussians and going through a similar splatting pipeline would exactly be transformed into 2D Gaussians. Our 2D Gaussians are coincidentally available for polarization priors. Thus, supervised by polarization priors, the learned 3D Gaussians capture more details, which represent the average orientation of normals by means and the changes within the neighborhood by covariance matrices.

## 3 METHODS

### 3.1 PRELIMINARY OF POLARIZATION

Here, we introduce the concept of polarization and its mathematical relation to surface normals projected to the captured images. The prior contributes to the disentanglement of specular radiance and geometry.

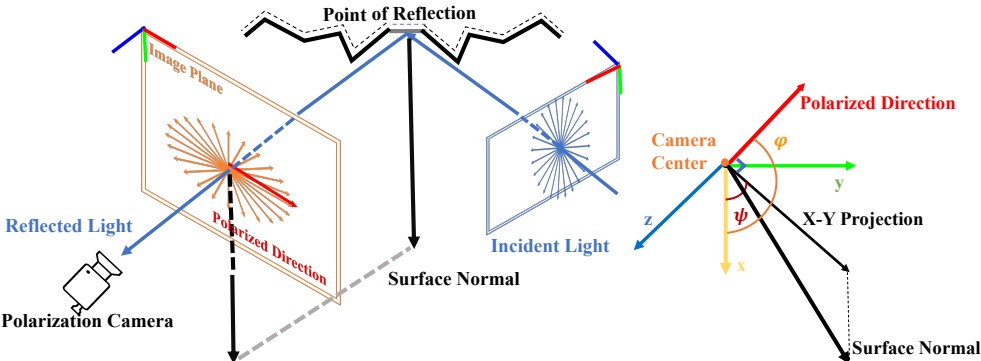

Figure 2: Illustration of polarization shift in specular reflection. The right figure is a detailed description of the geometry relation between AoP and the surface normal. $\psi$ is the azimuth angle. $\varphi$ is the AoP, which is the angle from the positive x-axis to the polarized direction.

Polarimetry describes the vibration status of light waves. Since light is a type of transverse wave that only oscillates in the plane perpendicular to the light path (Collett, 2005), the full polarimetric cues of rays are always represented by planar ellipses (Smith & Ward, 1974). The magnitude of vectors inside these ellipses alludes to the amplitude of the light wave vibration along the vectors, as shown in Fig. 2. Common light sources, such as sunlight and LED spotlights, emit unpolarized light, i.e., the light vibrates equally in all directions. In our captured scenes, objects are mostly illuminated directly by light sources, so we assume that the incident light is unpolarized. During reflection, the vibration in each direction is absorbed unequally, and unpolarized incident light turns into partially polarized reflected light captured by polarization cameras. The Angle of Polarization (AoP) and Degree of Polarization (DoP) are two cues of the polarization ellipse functionally related to projected surface normals at the points of reflection, which can be formulated as:

$$\boldsymbol{\varphi}(i,j) = \frac{1}{2} \arctan \frac{\mathbf{s}_2(i,j)}{\mathbf{s}_1(i,j)}, \ \boldsymbol{\rho}(i,j) = \frac{\sqrt{\mathbf{s}_1^2(i,j) + \mathbf{s}_2^2(i,j)}}{\mathbf{s}_0(i,j)}, \ \{\boldsymbol{\varphi}, \boldsymbol{\rho}\} \in \mathbb{R}^{H \times W}, \tag{1}$$

where $\boldsymbol{\varphi}, \boldsymbol{\rho}$ are AoP and DoP, $(i,j)$ is the pixel index, and $\mathbf{s} = [\mathbf{s}_0, \mathbf{s}_1, \mathbf{s}_2, \mathbf{s}_3]$ is Stokes vector directly calculated from polarization capture. Generally, in specularity-dominant regions, the relation between projected normals and AoP is fixed as Fig 2(b) and the equation $\psi + \frac{\pi}{2} \equiv \varphi \mod \pi$. Moreover, DoP is significantly higher in these regions. Details of polarization analysis are shown in the Appendix.

### 3.2 Gaussian Guided Polarimetric Neural 3D Reconstruction Pipeline

Polarimetric neural 3D reconstruction refers to reconstructing surfaces by neural implicit surface learning, given $N$ calibrated multi-view images $\mathcal{X} = \{\mathbf{C}_i\}_{i=1}^N$ with pixel-aligned polarization priors $\mathcal{Y} = \{\boldsymbol{\varphi}_i, \boldsymbol{\rho}_i\}_{i=1}^N$. First, we introduce a general pipeline of learning surface by volume rendering, taking NeuS (Wang et al., 2021) as an example. Sec. 3.2.2 introduces the 3D Gaussian of surface normals and its transforms to 2D Gaussian in the image plane. It represents the geometry of surface points more precisely and thus can separate detailed geometry from high-frequency specular radiance. 3.2.3 presents our full optimization containing radiance loss and Gaussian loss, which measures the gap between these 2D Gaussians and polarization priors. We propose a DoP reweighing strategy to alleviate the aforementioned noise and imbalanced distribution of polarization priors. It balances the influence of radiance and polarimetric cues adaptively. Finally, Sec. An overview of the entire framework is shown in Fig. 3.

#### 3.2.1 Learning Surface by Volume Rendering

NeRF (Mildenhall et al., 2020) proposed a novel render pipeline with a combination of spatial neural radiance fields and volume rendering (Kajiya & Von Herzen, 1984) to synthesize high-quality novel view images. Unlike traditional explicit meshes, the representation of 3D scenes in NeRF

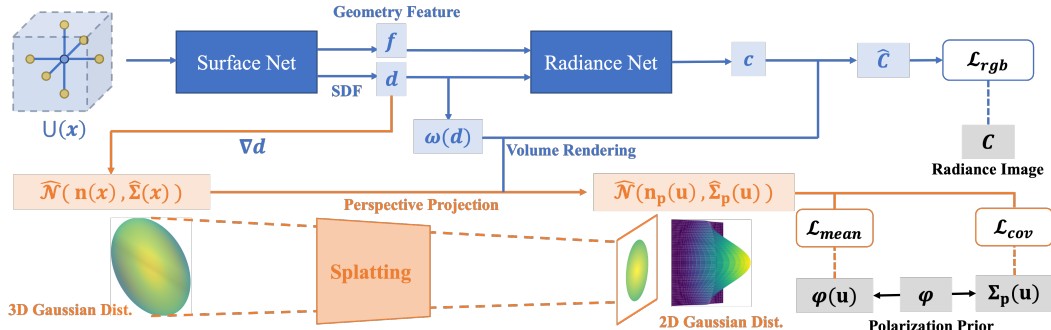

Figure 3: Illustration of our method.

is decomposed into spatial-dependent density fields and radiance fields depending on both spatial position and viewing direction. Then, the color of an arbitrary pixel with a ray $\mathbf{r} = \mathbf{o} + t\mathbf{d}$ passed through can be rendered by volume composition along the ray:

$$\hat{\mathbf{C}}(\mathbf{r}) = \sum_{k=1}^{K} T_i \alpha_i \mathbf{c}_i(\mathbf{r}_i, \mathbf{d}), \ T_i = \exp\left(-\sum_{j=1}^{i-1} \alpha_j \delta_j\right), \ \alpha_j = 1 - \exp\left(-\sigma_j(\mathbf{r}_j)\delta_j\right), \quad (2)$$

, where $K$ points $\{\mathbf{x}|\mathbf{o}+t_i\mathbf{d}\}_{i=1}^{K}$ on the ray are sampled. $\sigma_i$ and $\mathbf{c}_i$ are approximated volume density and radiance predicted by neural networks with position $\mathbf{x}$ and viewing direction $\mathbf{d}$ as inputs. $\delta_i$ is the length of sampled interval $[t_{i-1}, t_i]$. $\alpha_i$ and $T_i$ denote the transmittance and alpha value of points, and by them the final color is alpha composited (Max, 1995). The neural network is optimized by the mean square error between the ground truth color $\mathbf{C}(\mathbf{r})$ in the image and the rendered color $\hat{\mathbf{C}}(\mathbf{r})$.

Despite realistic novel view images, the geometry of scenes extracted from learned density fields is inaccurate with floating artifacts since the shape is not defined in the density field. NeuS (Wang et al., 2021) defines surfaces as the zero-level set of Signed Distance Field (SDF), and density is derived from SDF:

$$[d(\mathbf{x}_i), \mathbf{f}(\mathbf{x}_i)] = f(\mathbf{x}_i), \ \alpha_i = \max\left(\frac{\Phi_s\left(d\left(\mathbf{x}_i\right)\right) - \Phi_s\left(d\left(\mathbf{x}_{i+1}\right)\right)}{\Phi_s\left(d\left(\mathbf{x}_i\right)\right)}, 0\right), \ \mathbf{c}_i = c(\mathbf{x}_i, \mathbf{n}_i, \mathbf{d}, \mathbf{f}_i), \quad (3)$$

where $c$ and $f$ are the geometry network and radiance network, $d(\mathbf{x}_i)$ is the signed distance to the surface and $\mathbf{f}_i = \mathbf{f}(\mathbf{x}_i)$ is the geometry feature. $\alpha_i$ is defined by SDF with Laplace distribution $\Phi_s(x) = (1 + e^{-sx})^{-1}$, where the variance $s$ is a trainable parameter. The volume rendering process is analogous to NeRF, while the radiance network takes normals $\mathbf{n}_i = \nabla_{\mathbf{x}} d(\mathbf{x}_i)$ and geometry feature $\mathbf{f}_i$ as additional inputs.

### 3.2.2 GAUSSIAN SPLATTING OF NORMALS

Neural SDF-based 3D reconstruction excels at smooth Lambertian objects. With neural BRDF defining the specular reflection between rays and surfaces, smooth surfaces of reflective objects can also be properly learned. However, the low-frequency implicit bias of neural networks (Xu et al., 2019) is a barrier for both of them to recover delicate geometry behind specular reflection, such as abrupt normal changes in NeRO (Liu et al., 2023). Thus, we propose a 3D Gaussian estimation of distributions of normals as an additional representation of geometry details. We show how it can be splatted to the image plane, making it available for 2D polarization supervision.

Instead of separate vectors assigned to each point, the normal within the neighborhood of an arbitrary position $\mathbf{x_i}$ is assumed as a Gaussian:

$$\mathcal{G}(\mathbf{x}|\mathbf{x}_i) = \mathcal{N}(\mathbf{n}(\mathbf{x}_i), \mathbf{\Sigma}(\mathbf{x}_i)) = \frac{1}{(2\pi)^{\frac{3}{2}}|\mathbf{\Sigma}(\mathbf{x}_i)|^{\frac{1}{2}}} \exp\left(-\frac{1}{2}(\mathbf{x} - \mathbf{n}(\mathbf{x_i}))^{\mathrm{T}}\mathbf{\Sigma}(\mathbf{x_i})^{-1}(\mathbf{x} - \mathbf{n}(\mathbf{x_i}))\right), z$$

$$(4)$$

where $\mathbf{n} \in \mathbb{R}^3$ is the normal, and $\mathbf{\Sigma} \in \mathbb{R}^{3\times3}$ is the covariance of the Gaussian. Given a ray with discretization $\{\mathbf{x}_i|\mathbf{x} + t_i\mathbf{d}\}_{k=1}^{K}$, additional $M$ positions within the neighborhood are super-sampled

to estimate the covariance. In this paper, $M$ is 6 containing $\mathbf{x}_{i-1}$, $\mathbf{x}_{i+1}$ and four positions around the ray. Hence, the unbiased estimation of Gaussian can be formulated as:

$$\hat{\mathcal{G}}(\mathbf{x}|\mathbf{x}_i) = \mathcal{N}(\mathbf{n}(\mathbf{x}_i), \hat{\boldsymbol{\Sigma}}(\mathbf{x}_i)) = \mathcal{N}\left(\mathbf{n}(\mathbf{x}_i), \frac{1}{M-1}\sum_{j=1}^{M}\left(\mathbf{n}(\mathbf{x}_i^j) - \mathbf{n}(\mathbf{x}_i)\right)\left(\mathbf{n}(\mathbf{x}_i^j) - \mathbf{n}(\mathbf{x}_i)\right)^{\mathrm{T}}\right),$$

(5)

where $\mathbf{n}(\mathbf{x}_i^j) = \nabla_{\mathbf{x}}d(\mathbf{x}_i^j), \mathbf{n}(\mathbf{x}_i^j) \in \mathbb{R}^3$. However, those 3D Gaussians are not accessible in captured 2D images, and volume rendering in Sec. 3.2.1 only takes 3D scalar fields into account, making the projection of 3D Gaussians non-trivial. Alternatively, (Zwicker et al., 2001) presents a splatting approach treating colors in 3D space as Gaussian kernels and visualizing them on the image plane. We apply analogous transforms and further prove our normal-based 3D Gaussians are exactly splatted to 2D Gaussians. Given a viewpoint, the transform can be formulated as:

$$\hat{\mathcal{G}}(\mathbf{x}|\mathbf{x}_i)_{\mathbf{P}} = \mathcal{N}(\mathbf{JW}\mathbf{n}(\mathbf{x}_i), \mathbf{JW}\hat{\boldsymbol{\Sigma}}(\mathbf{x}_i))\mathbf{W}^{\mathrm{T}}\mathbf{J}^{\mathrm{T}}) = \mathcal{N}\left(\begin{bmatrix}\mathbf{n}_{\mathbf{p}}(\mathbf{x}_i)\\0\end{bmatrix}, \begin{bmatrix}\hat{\boldsymbol{\Sigma}}_{\mathbf{p}}(\mathbf{x}_i) & \\ & 0\end{bmatrix}\right), \quad (6)$$

where $\mathbf{W} \in \mathbb{R}^{3\times 3}$, $\mathbf{J} \in \mathbb{R}^{3\times 3}$ are viewing transform matrix and normal projection matrix (Chen et al., 2022), respectively. Derivation of them is shown in the Appendix. It shows that only the first two rows of the transformed mean vector and the upper $2 \times 2$ square block of the transformed covariance matrix remain non-zero, splatting 3D Gaussians to 2D Gaussians in the image plane. For simplification, 2D Gaussians are also denoted by $\hat{\mathcal{G}}(\mathbf{x}|\mathbf{x}_i)_{\mathbf{P}} = \mathcal{N}(\mathbf{n}_{\mathbf{p}}(\mathbf{x}_i), \hat{\boldsymbol{\Sigma}}_{\mathbf{p}}(\mathbf{x}_i))$, where $\mathbf{n}_{\mathbf{p}} \in \mathbb{R}^2, \hat{\boldsymbol{\Sigma}}_{\mathbf{p}}(\mathbf{x}_i)) \in \mathbb{R}^{2\times 2}$. Moreover, the SVD of the covariance matrix $\hat{\boldsymbol{\Sigma}}_{\mathbf{P}} = \hat{\mathbf{V}}\hat{\boldsymbol{\Lambda}}\hat{\mathbf{V}}^{\mathrm{T}}$ circumvents the ill-posedness of the covariance matrix and reveals its relation to anisotropic normal distribution. Intuitively, if the geometry appears smooth from the imaging perspective, then the corresponding normals of the neighborhood will be projected to similar vectors, resulting in an insignificant deviation of the eigenvalues. Otherwise, the deviation would be significant. Eigenvectors also show the local shape of the position, as shown in Fig. 4 (e). Finally, all 2D Gaussians on the ray passing through the pixel $\mathbf{u}$ is composited by volume rendering:

$$\hat{\mathcal{G}}(\mathbf{u}) = \mathcal{N}\left(\sum_{k=1}^{K}T_i\alpha_i\mathbf{n}_{\mathbf{p}}(\mathbf{x}_i), \sum_{k=1}^{K}T_i\alpha_i\hat{\boldsymbol{\Sigma}}_{\mathbf{p}}(\mathbf{x}_i)\right) = \mathcal{N}(\mathbf{n}_{\mathbf{p}}(\mathbf{u}), \hat{\boldsymbol{\Sigma}}_{\mathbf{p}}(\mathbf{u})),$$

(7)

where $T_i$ and $\alpha_i$ are in Eq. 2. $\mathbf{n}_{\mathbf{p}}(\mathbf{u}) \in \mathbb{R}^2, \hat{\boldsymbol{\Sigma}}_{\mathbf{p}}(\mathbf{u}) \in \mathbb{R}^{2\times 2}$. The mean of 3D Gaussians $\mathbf{n}(\mathbf{x}_i)$, which is splatted to $\mathbf{n}_{\mathbf{p}}(\mathbf{u})$, represents the overall orientation of $\mathbf{x}_i$. And the covariance $\hat{\boldsymbol{\Sigma}}(\mathbf{x}_i))$ and splatted $\hat{\boldsymbol{\Sigma}}_{\mathbf{p}}(\mathbf{u}))$ in the image model the high-frequency details. In this way, our representation captures more details than NeuS and other methods based on the neural BRDF parameterized by isotropic normals distribution. Another main strength of those 2D Gaussians is direct supervision by polarization priors, which is introduced in Sec. 3.2.3.

### 3.2.3 OPTIMIZATION WITH REWEIGHTED POLARIZATION PRIORS

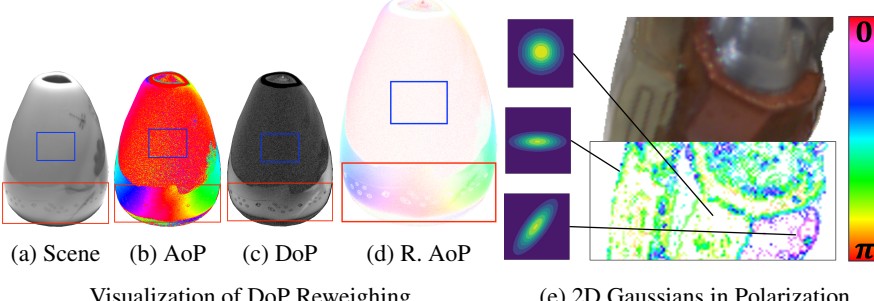

(a) Scene   (b) AoP   (c) DoP   (d) R. AoP

Visualization of DoP Reweighing    (e) 2D Gaussians in Polarization

Figure 4: Visualization of Reweighted AoP Priors. Red boxes bound specular reflection dominant regions, and the blue boxes bound diffuse ones. (d) is the AoP map reweighted by DoP. Saturation in (e) indicates the degree of anisotropy, and color represents the direction of the singular vector of 2D Gaussians' covariance. A few 2D Gaussians are drawn as ellipses for intuition.

The 2D Gaussian in Sec. 3.2.2 can be directly extracted from AoP priors $\{\boldsymbol{\varphi}_i\}_{i=1}^{N}$ in Eq. 1 by:

$$\widetilde{\mathcal{G}}_{\mathbf{p}}(\mathbf{u}|\mathbf{u}_i) = \mathcal{N}(\widetilde{\mathbf{n}}_{\mathbf{p}}(\mathbf{u}_i), \widetilde{\boldsymbol{\Sigma}}_{\mathbf{p}}(\mathbf{u}_i))$$

$$= \mathcal{N}\left(s\mathbf{v}(\boldsymbol{\psi}(\mathbf{u}_i)), \frac{1}{M'-1}\sum_{j=1}^{M'}\left(\mathbf{v}(\boldsymbol{\psi}(\mathbf{u}_i^j)) - \mathbf{v}(\boldsymbol{\psi}(\mathbf{u}_i)))\right)\left(\mathbf{v}(\boldsymbol{\psi}(\mathbf{u}_i^j)) - \mathbf{v}(\boldsymbol{\psi}(\mathbf{u}_i)))\right)^{\mathrm{T}}\right),$$

(8)

where $\mathbf{u}_i^{(j)} = (\mathbf{x}_i^{(j)})_{\mathbf{p}}$ is the corresponding pixel index of (super-sampled) points on the ray. Therefore, $M' = 4$ since $\mathbf{x}_{i-1}$ and $\mathbf{x}_{i+1}$ are on the same ray as $\mathbf{x}_i$. $\mathbf{v}(\theta)$ represents a 2D unitary vector rotated by $\theta$. $\boldsymbol{\Psi} \equiv \boldsymbol{\varphi} + \frac{\pi}{2} \mod \pi$ is the azimuth angle of normals, derived from the AoP in Eq. 1. And $s$ is a scale factor. Similar to Sec. 3.2.2, the estimated covariance matrix is decomposed into $\widetilde{\boldsymbol{\Sigma}} = \widetilde{\mathbf{V}}\widetilde{\boldsymbol{\Lambda}}\widetilde{\mathbf{V}}^{\mathrm{T}}$. We define the degree of anisotropy (DoA) of those 2D Gaussians as $\frac{\boldsymbol{\Lambda_0}}{\boldsymbol{\Lambda_1}}$. 2D Gaussians saturated by DoA are visualized in Fig. 4 (e). In this Fig, color is concentrated to and coherent along the edges of the scene. It shows DoA is higher in the region with complicated geometry and surface changes most dramatically along singular vectors of covariance. Before optimization, the polarization prior AoP is reweighted by DoP to alleviate the aforementioned observational noise and imbalanced distribution problem in Sec. 1. The noise of AoP is mainly generated by diffuse reflection because it's always weakly polarized (Kajiyama et al., 2023). The DoP in diffuse-dominant regions is significantly lower than specular-dominant ones, as shown in Fig. 4 (c) and (d). Thus, the reweighted AoP defined as $\boldsymbol{\varphi} \cdot \boldsymbol{\rho}$ is proposed as an alternative supervision with less noise. Meanwhile, radiance is disentangled with surroundings in specular reflection dominant areas. To adaptively balance radiance and polarization priors, our full loss function during reconstructing is defined as:

$$\mathcal{L} = \alpha(1-\boldsymbol{\rho})\mathcal{L}_{\text{color}} + \beta\boldsymbol{\rho}(\mathcal{L}_{\text{mean}} + \mathcal{L}_{\text{cov}}) + \gamma\mathcal{L}_{\text{eik}} + \delta\mathcal{L}_{\text{mask}},$$

$$\mathcal{L}_{\text{color}} = \| \hat{\mathbf{C}}(\mathbf{u}) - \mathbf{C}(\mathbf{u}) \|_2, \ \mathcal{L}_{\text{mean}} = \| \hat{\boldsymbol{\varphi}}(\mathbf{n}_{\mathbf{p}}(\mathbf{u})) - \boldsymbol{\varphi}(\mathbf{u}) \|_1,$$

$$\mathcal{L}_{\text{cov}} = \left(\left\|\frac{\hat{\boldsymbol{\Lambda}}_1}{\hat{\boldsymbol{\Lambda}}_0} - \frac{\widetilde{\boldsymbol{\Lambda}}_1}{\widetilde{\boldsymbol{\Lambda}}_0}\right\|_1 + \beta' < \hat{\mathbf{V}}, \widetilde{\mathbf{V}} >\right)(\mathbf{u}), \ \mathcal{L}_{\text{eik}} = \frac{1}{K}\sum_{i=1}^{K}(\|\nabla_{\mathbf{x}}d(\mathbf{x}_i)\|_2 - 1)^2,$$

(9)

where $\mathcal{L}_{\text{color}}$ and $\mathcal{L}_{\text{mask}}$ are the radiance rendering loss and the BCE loss of object masks in NeuS (Wang et al., 2021). Splatted 2D Gaussians is supervised by $\boldsymbol{\psi}(\mathbf{u})$ and $\widetilde{\boldsymbol{\Sigma}}_{\mathbf{p}}(\mathbf{u})$ in Eq. 8. $\hat{\boldsymbol{\varphi}}(\mathbf{n}_{\mathbf{p}}(\mathbf{u})) \equiv \boldsymbol{\psi}(\mathbf{u}) + \frac{\pi}{2} \mod \pi$ and $\boldsymbol{\psi}$ is the azimuth angle of normals. The supervision of radiance and polarization priors are reweighted by the DoP $\boldsymbol{\rho}$. Especially, only Anisotropy (ratio of singular values) and eigenvectors are supervised to avoid scaling and numerical issues. If the local shape is like a plane, normals will change smoothly in all directions, and the Anisotropy approaches 1. If there are some details like edges, normals tend to change abruptly and exhibit directionality, represented by eigenvectors. $\mathcal{L}_{\text{eik}}$ is a regularization term of the gradient of SDF widely used (Gropp et al., 2020). $\alpha, \beta, \gamma$ and $\delta$ are hyper-parameters.

## 4    EXPERIMENTS

To evaluate the effectiveness of our method, we tested GNeRP on objects from multiple scenes and compared them with existing state-of-the-art neural 3D reconstruction methods.

**PolRef Dataset**    The methods are evaluated on the PANDORA dataset (Dave et al., 2022) and captured scenes by ourselves. The PANDORA includes 3 reflective objects (Owl, Blackvase, and Gnome) with polarization priors. However, their ground truth shapes are unavailable for quantitative evaluation. Moreover, the diversity of materials, geometry, and illumination is not enough for overall comparisons. Only the geometry of the Gnome scene is complicated but less reflective. Only a mirror-like ball in the Blackvase reflects surroundings other than highlights. Other common datasets, including Shiny Blender (Verbin et al., 2022), lack polarization priors for our method. To comprehensively evaluate the performance of 3D reconstruction methods, a new challenging multi-view dataset named PolRef was collected, consisting of objects with reflective and less-textured surfaces captured with various illumination. Radiance images and aligned polarization priors were captured in one shot using polarization cameras. To obtain precise and complete ground truth shapes, objects were produced using SLA 3D printers, with an accuracy tolerance of $\pm 0.1mm$. Detailed descrip-

tions are shown in the Appendix. The dataset will be released to facilitate further research on 3D reconstruction in more challenging scenes in the future.

**Experimental Settings**   GNeRP is built upon NeuS (Wang et al., 2021). The geometry network and radiance network in Fig. 3 is the same as that of NeuS. Since the covariance loss $\mathcal{L}_{\text{cov}}$ in Eq. 9 refines the details of the geometry, it will not be activated during the initial $50K$ steps. The model is trained for 200k iterations and takes about 6 hours on a server with 4 NVIDIA RTX 3090 Ti GPUs for the reconstruction. After optimization, the meshes are extracted from learned SDF by Marching Cubes (Lorensen & Cline, 1998) with a resolution of $512^3$. The hyper-parameter settings are shown in the Appendix D.3.

## 4.1   COMPARISON WITH STATE-OF-THE-ART METHODS

We conducted the comparison of reconstruction accuracy between our methods and several state-of-the-art methods, including baseline methods for neural 3D reconstruction (Unisurf (Oechsle et al., 2021), VolSDF (Yariv et al., 2021), and NeuS (Wang et al., 2021)), two view-consistency based methods (NeuralWarp (Darmon et al., 2022) and Geo-NeuS (Fu et al., 2022)), two new methods for reconstruction of reflective objects (NeRO (Liu et al., 2023) and Ref-NeuS (Ge et al., 2023)), and a polarization-based method (PANDORA (Dave et al., 2022)).

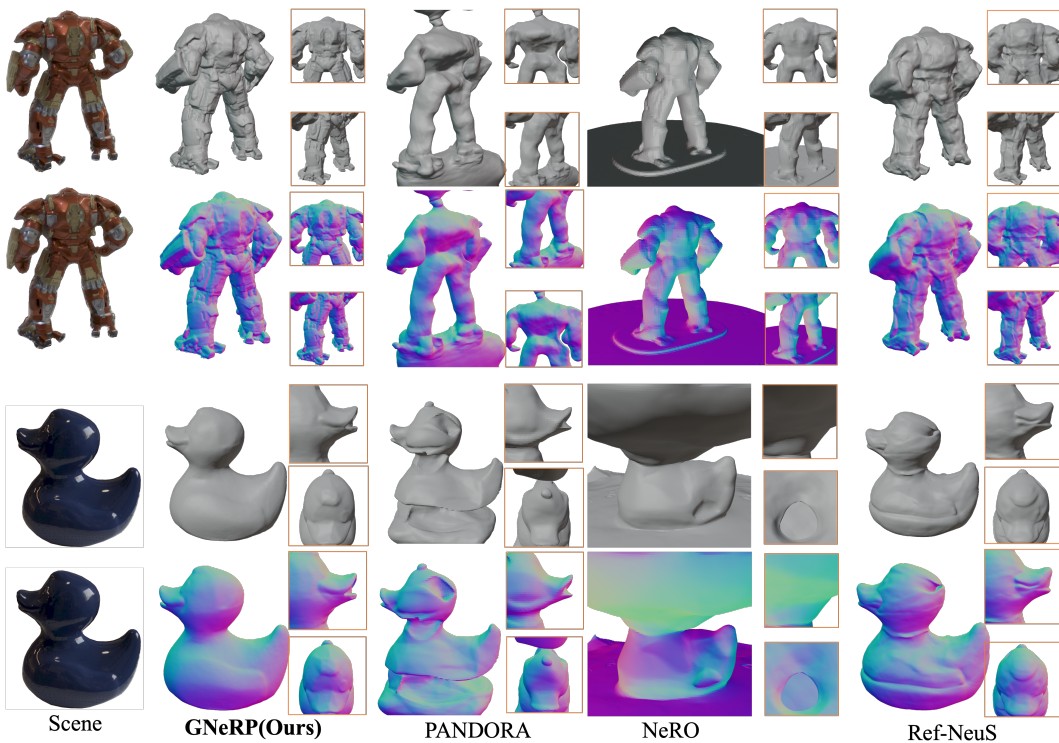

| Scene | **GNeRP(Ours)** | PANDORA | NeRO | Ref-NeuS |

Figure 5: Visual comparison of our method and state-of-the-art methods.

A qualitative comparison between our method and state-of-the-art methods specially designed for reflective objects is shown in Fig. 7, which demonstrates that our method significantly improves the geometry details and accuracy of normals. In the Ironman scene, NeRO reconstructed an over-smoothed geometry. Due to the spatial continuity of neural BRDF, it failed to reconstruct the high-frequency armor details with abrupt normal changes. The shape of Ref-NeuS is more accurate, but the sole scalar SDF is not able to predict the geometry details. The duck scene is more reflective with a combination of highlights and reflection of surroundings. Although Ref-NeuS detected the reflective regions, it was still misled by the environment radiance and reconstructing concave holes. The results of PANDORA are over-smoothed in Ironman and disturbed by noise in polarization priors in Duck. Additional comparisons of different scenes are shown in the Appendix. We conduct

| Methods | Ironman | Duck | Cow | Snorlax | Mean |
|---|---|---|---|---|---|
| Unisurf* (Oechsle et al., 2021) | 3.97 | 10.83 | 14.43 | 14.33 | 10.89 |
| VolSDF (Yariv et al., 2021) | 2.72 | 5.16 | 5.95 | 3.20 | 4.26 |
| NeuS (Wang et al., 2021) | 2.28 | 2.12 | 3.82 | 2.11 | 2.58 |
| Geo-NeuS (Fu et al., 2022) | 4.77 | 10.12 | 17.48 | 10.39 | 10.82 |
| NeuralWarp (Darmon et al., 2022) | 12.44 | 19.78 | 5.41 | 20.57 | 14.55 |
| NeRO* (Liu et al., 2023) | 2.29 | 23.75 | 2.95 | 24.30 | 13.32 |
| Ref-NeuS (Ge et al., 2023) | 1.88 | 1.93 | 3.66 | 1.99 | 2.34 |
| PANDORA $^\dagger$ (Dave et al., 2022) | 4.61 | 5.28 | 7.96 | 5.73 | 5.90 |
| GNeRP | **1.34** | **1.63** | **1.39** | **1.05** | **1.35** |

Table 1: Quantitative comparison with state-of-the-art methods. The lower is better. * indicates the method doesn't use object masks. † refers to the use of polarization priors. The best scores are **bold**, the second best scores are double underlined, and the third best scores are underlined.

quantitative comparisons on the four scenes with ground truth meshes in our dataset. The evaluation metric is Chamfer Distance according to NeuS (Wang et al., 2021) and Unisurf (Oechsle et al., 2021). Scores are reported in Table 1, which shows our method reconstructs more precise meshes in all four scenes. NeRO (Liu et al., 2023) needs environment information to calculate occlusion loss, and Unisurf also learns occupancy from backgrounds. Training them with masks failed directly, so we report the scores without masks in Tab. 1 denoted by *. Geo-NeuS needs sparse points from Structure-from-Motion (SfM) Schönberger et al. (2016) to calculate SDF loss and select the pairs based on SfM for the warping process. We did the sparse reconstruction in COLMAP and followed the pairs selection method in NeuralWarp. Full polarimetric acquisition (Stokes vector $[\mathbf{s}_0, \mathbf{s}_1, \mathbf{s}_2]$, see in the Appendix) is required by PANDORA. We processed the raw polarization capture to follow its data conventions. Sparse reconstruction of reflective scenes was noisy and incomplete, resulting in the worst accuracy by Geo-NeuS and NeuralWarp. Ref-NeuS demonstrated comparable scores on all scenes, but our method still outperformed.

## 4.2 ABLATION STUDY

| Scene | NeuS | w/ $\mathcal{L}_{\mathrm{mean}}$ | w/ $\mathcal{L}_{\mathrm{cov}}$ | w/ ReW. $\mathcal{L}_{\mathrm{mean}}$ | w/ ReW. $\mathcal{L}_{\mathrm{cov}}$ | Full |
|---|---|---|---|---|---|---|
| Snorlax | 2.11 | 2.03 | 3.01 | 1.81 | 2.07 | **1.05** |
| Cow | 3.82 | 2.72 | 5.54 | 1.94 | 2.29 | **1.39** |

Table 2: Ablation Study. $\mathcal{L}_{\mathrm{mean}}$, $\mathcal{L}_{\mathrm{cov}}$ are in Eq. 9.

To validate the effectiveness of the proposed modules, we test the following three settings as shown in Tab. 2. W/$\mathcal{L}_{\mathrm{mean}}$ refers to the naive supervision of $\varphi$ and azimuth angle of normals in SDF. Due to the noise, the results are worse. W/$\mathcal{L}_{\mathrm{mean}}$ refers to the polarization supervision with only covariance. The reconstruction is focused on details and results in the worst scores. W/ ReW. $\mathcal{L}_{\mathrm{mean}}$ indicates the reweighted losses $(1 - \rho)\mathcal{L}_{\mathrm{color}} + \rho\mathcal{L}_{\mathrm{mean}}$. Similarly, w/ReW. $\mathcal{L}_{\mathrm{cov}}$ represents $(1 - \rho)\mathcal{L}_{\mathrm{color}} + \rho\mathcal{L}_{\mathrm{cov}}$. The reweighting does improve the efficiency of polarization priors. Finally, the full setting shows the best scores. Additional visualization is shown in the Appendix.

## 5 CONCLUSION

We propose GNeRP to reconstruct the detailed geometry of reflective scenes. In GNeRP, we propose a new Gaussian-based representation of normals and introduce polarization priors to supervise it. We propose a DoP reweighing strategy to resolve noise issues in polarization priors. We collect a new, challenging multi-view dataset with non-Lambertian scenes to evaluate existing methods more comprehensively. Experimental results demonstrate the superiority of our method.

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

## A  METHOD DETAILS

### A.1  POLARIZATION ANALYSIS

The primal polarization information captured in one shot is a four-directional polarization image obtained by using four directional linear polarizers at angles of $0°$, $45°$, $90°$, and $135°$. After the demosaicing process, the four directional images can be denoted as $\mathbf{I}_0$, $\mathbf{I}_{45}$, $\mathbf{I}_{90}$, and $\mathbf{I}_{135}$ respectively. For computational efficiency, the Stokes representation is widely used, such as in PANDORA Dave et al. (2022), which is defined as follows:

$$\mathbf{S} = \begin{bmatrix} \mathbf{s}_0 \\ \mathbf{s}_1 \\ \mathbf{s}_2 \\ \mathbf{s}_3 \end{bmatrix} = \begin{bmatrix} \mathbf{I}_0 + \mathbf{I}_{90} \\ \mathbf{I}_0 - \mathbf{I}_{90} \\ \mathbf{I}_{45} - \mathbf{I}_{135} \\ \mathbf{0} \end{bmatrix}, \mathbf{I} \in \mathbb{R}^{H \times W \times 3}. \tag{10}$$

It is noted that the fourth component of Stokes vector is zero because it represents circular polarization, which is not captured by linear polarizers. The vector also covers radiance information, which can be derived as:

$$\mathbf{I}_{\mathrm{rad}} = \frac{1}{2}\mathbf{s}_0. \tag{11}$$

However, the Stokes vector is always noisy and redundant for 3D reconstruction. This is because, in addition to the orientation of surfaces, the Stokes vector is affected by environmental variables and the nature of objects. These variables include, but are not limited to, illumination, roughness, and material properties. PANDORA Dave et al. (2022) supervises the full vector for inverse rendering, but it's unnecessary for learning geometry only. Therefore, we refine the Stokes vector into the AoP and DoP cues, as they are most related to the geometry within the Stokes vector. Calculating AoP and DoP from Stokes vector can be formulated as follows:

$$\begin{aligned} \boldsymbol{\varphi}(i,j) &= \frac{1}{2}\arctan\frac{\mathbf{s}_2(i,j)}{\mathbf{s}_1(i,j)} \\ \boldsymbol{\rho}(i,j) &= \frac{\sqrt{\mathbf{s}_1(i,j)^2 + \mathbf{s}_2(i,j)^2}}{\mathbf{s}_0(i,j)} \end{aligned}, \{\boldsymbol{\varphi}, \boldsymbol{\rho}\} \in \mathbb{R}^{H \times W}, \tag{12}$$

where $\boldsymbol{\varphi}(i,j)$ is the AoP at the pixel $(i,j)$, lying in the interval of $[0, \pi]$. And $\boldsymbol{\rho}(i,j)$ is the DoP ranging from 0 to 1.

### A.2  PERSPECTIVE TRANSFORM OF GAUSSIANS

Similar to (Zwicker et al., 2001), transforms of 3D Gaussians to 2D Gaussians are performed based on the lemma:

**Lemma 1** *Given a 3D affine transform* $\mathbf{u} = \boldsymbol{\Phi}(\mathbf{x}) = \mathbf{M}\mathbf{x} + \mathbf{c}$, *a 3D Gaussian* $\mathcal{G}(\mathbf{x}) = \mathcal{N}(\boldsymbol{\mu}, \boldsymbol{\Sigma})$ *is transformed into:*

$$\mathcal{G}'(\mathbf{u}) = \mathcal{N}(\boldsymbol{\Phi}(\boldsymbol{\mu}), \mathbf{M}\boldsymbol{\Sigma}\mathbf{M}^T) \tag{13}$$

*It can be proven by replacing* $\mathbf{x}$ *with* $\boldsymbol{\Phi}^{-1}(\mathbf{u})$.

During the imaging, points in the world space are projected to 2D pixels through the view transform matrix and perspective projection matrix. The view transform converts the world coordinates to the camera ones. It can be formulated as $\mathbf{\Phi}(\mathbf{x}) = \mathbf{R}\mathbf{x} + \mathbf{t}$, where $\mathbf{R}$ is a rotation matrix and $\mathbf{t}$ is a translation vector. For bound normals, the translation will not change their orientation, and thus the corresponding matrix $\mathbf{W} = \mathbf{R}$. However, the perspective project matrix of normals is quite different from points. For 3D points, the transform is non-linear projecting points to the image plane. On the contrary, the projection of normals is proven to be linear, which can be formulated as (Chen et al., 2022):

$$\mathbf{z} \times (\mathbf{v} \times \mathbf{n}) = [-v_z n_x + v_x n_z, -v_z n_y + v_y n_z, 0]^T = \begin{bmatrix} -v_z & 0 & v_x \\ 0 & -v_z & v_y \\ 0 & 0 & 0 \end{bmatrix} \mathbf{n} = \mathbf{J}\mathbf{n}, \quad (14)$$

where $\mathbf{n}$ is the normal vector, $\mathbf{v}$ is the vector points to the location of the normal vector, i.e., the direction of the ray, and $\mathbf{z} = [0, 0, 1]^T$. Since both of the view transform and projection in Eq. 14 are affine, the structure of Gaussian holds. Moreover, the last row of $\mathbf{J}$ is all-zero, making only the upper $2 \times 2$ square block of $\mathbf{J}\mathbf{M}\mathbf{J}^\mathbf{T}$ and the first two rows of $\mathbf{J}\mathbf{x}$ are non-zero. It explains the last term in Eq. 6.

### A.3 Gaussians Estimation and Decomposition

Given a normal vector of a 3D point $\mathbf{n}(\mathbf{x}_i)$ with normal vectors of super-sampled points with the neighborhood $\{\mathbf{n}(\mathbf{x}_i^j)\}_{j=1}^M$, a 3D Gaussian $\mathcal{N}(\mathbf{n}(\mathbf{x}_i), \hat{\mathbf{\Sigma}}(\mathbf{x}_i))$ can be estimated by Eq.5. Then, the 3D Gaussian is transformed into the camera coordinates $\mathcal{N}(\mathbf{n_p}(\mathbf{x}_i), \hat{\mathbf{\Sigma}}_\mathbf{p}(\mathbf{x}_i))$ by Eq. 6. Finally, all 3D Gaussians along the same ray are composited into a 2D Gaussian $\mathcal{N}(\mathbf{n_p}(\mathbf{u}), \hat{\mathbf{\Sigma}}_\mathbf{p}(\mathbf{u}))$ in the pixel $\mathbf{u}$ by volume rendering in Eq. 7, where $\mathbf{x}_i = \mathbf{o}(\mathbf{u}) + t_i \mathbf{d}(\mathbf{u})$. Then Singular Value Decomposition (SVD) (Klema & Laub, 1980) is performed to get $\hat{\mathbf{\Lambda}}(\mathbf{u})$:

$$\hat{\mathbf{\Sigma}}_\mathbf{p}(\mathbf{u}) = \hat{\mathbf{V}}\hat{\mathbf{\Lambda}}\hat{\mathbf{V}}^\mathrm{T}(\mathbf{u}) \quad (15)$$

For 2D polarization priors, 2D Gaussians are estimated by Eq. 8. For the sample $\mathbf{x}_i^j$, the corresponding 2D pixel can be located by the ray $\mathbf{x}_i^j = \mathbf{o} + t_i \mathbf{d}^j$ since $\mathbf{d}^j$ and the pixel $\mathbf{u}^j$ is bijective. From the AoP $\varphi(\mathbf{u})$, we can derive the orientation of the projected normal vector $\psi$. It's represented by:

$$s \cdot \mathbf{v}(\boldsymbol{\psi}(\mathbf{u}^j)) = [\cos(\psi(\mathbf{u}^j)), \sin(\psi(\mathbf{u}^j)]^T, \ \psi(\mathbf{u}^j) = \left(\varphi(\mathbf{u}^j) + \frac{\pi}{2}\right)_{\mod \pi}, \quad (16)$$

where $s$ is the scale factor equal to the magnitude of the projected normal vector. Through Eq. 8, 2D Gaussians of polarization images can be estimated. Finally the same SVD is performed to get $\widetilde{\mathbf{\Lambda}}$.

## B POLREF DATASET

Since ground truth geometry is inaccessible in the most of existing polarimetric multi-view datasets, we sampled a new dataset named PolRef Dataset to evaluate our method comprehensively. The dataset is split into real captured scenes and synthetic scenes.

**Dataset Collection** The dataset consists of 8 scenes (Ironman, Snorlax, Duck, Cow, Cat, Vase, Bunny, and Dragon). For real scenes, the capture pipeline is illustrated in Fig. 6 (a). Radiance images and aligned polarimetric priors were captured in one shot using polarization cameras (LUCID PHX050S-Q and HIKIVISION MV-CH050-10UP). We captured multiple views around the object. The objects were put on a calibration disk designed for $360°$ capture to get poses. The data processing formulation to extract polarization priors is listed in Sec. A.1. To obtain precise and complete ground truth shapes, 4 objects (Ironman, Snorlax, Duck, and Cow) were produced using SLA 3D printers, with an accuracy tolerance of $\pm 0.1mm$, given STL files as ground truth shapes. To enhance the diversity of the dataset, two non-3D printed objects were also included in the data collection process. Moreover, to increase the diversity and evaluate our method more comprehensively, synthetic data (Bunny and Dragon) is generated using the Mitsuba renderer (Nimier-David et al., 2019). Mitsuba is able to render polarization priors (Stokes vectors) from meshes with pre-defined attributes of scenes, as illustrated in Fig. 6 (b). In addition to polarization priors, ground truth normal maps

are accessible through rendering, making the evaluation of normals accuracy available. The dataset will be released to facilitate further research on 3D reconstruction in more challenging scenes in the future.

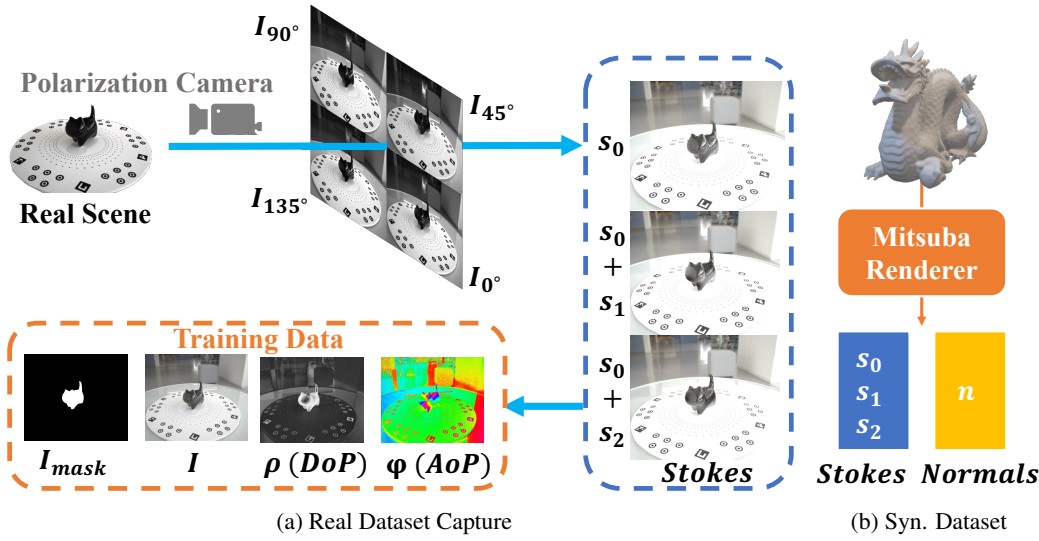

(a) Real Dataset Capture

(b) Syn. Dataset

Figure 6: Pipeline of PolRef Dataset.

**Evaluation Protocol** For real and synthetic datasets, the accuracy of reconstructed meshes is measured by Chamfer Distance (CD), which can be formulated as:

$$\mathrm{CD}\left(P_1, P_2\right) = \frac{1}{2n} \sum_{i=1}^{n} \left|x_i - \mathrm{Nearest}\left(x_i, P_2\right)\right| + \frac{1}{2m} \sum_{j=1}^{n} \left|x_j - \mathrm{Nearest}\left(x_j, P_1\right)\right|, \qquad (17)$$

where $x_i$ is the vertex, $n, m$ are the number of vertices in meshes, and $\mathrm{Nearest}(x, P) = \mathrm{argmin}_{x' \in P} \|x - x'\|$ is nearest neighboring function. Mean Angular Error (MAE) of normals is introduced to the evaluation of synthetic data since ground truth normals are available, which can be formulated as:

$$\mathrm{MAE}\left(\hat{\mathbf{n}}, \mathbf{n}\right) = \frac{1}{NM} \arccos\left(\frac{\hat{\mathbf{n}}_i^j \cdot \mathbf{n}_i^j}{\|\hat{\mathbf{n}}_i^j\|\|\mathbf{n}_i^j\|}\right), \qquad (18)$$

where $\mathbf{n}_i^j$ is the ground truth normal vector at the pixel $i$ in the view $j$, and $N$, $M$ are the number of views and pixels, respectively. Compared to CD, MAE is more sensitive to details of shapes, where normals change abruptly.

## C ADDITIONAL COMPARISONS WITH EXISTING METHODS

### C.1 VISUAL COMPARISON ON PANDORA DATASET

Visual comparison is shown in Fig. 7. In the Owl scene, all baseline methods fail to reconstruct feathers, of which surfaces are mistakenly concave. Because baseline methods recognize dark radiance as the cue of deeper surface points. In the Black Vase and Cat scenes, Unisurf and NeuS suffer from severe shape-radiance ambiguity as they cannot disentangle specular reflection with surface color and wrongly reconstruct the shapes of reflected scenes. VolSDF distinguishes reflection and color more clearly than them but still worse than our method in glossy areas, as shown in the Cat scene. In the Vase scene, even though we have tried as many combinations of hyper-parameters as we can, Unisurf still fails to reconstruct rough geometry. VolSDF isn't able to find correct surface points from dark radiance, as the hole of the vase is wrongly recognized as a convex surface. The reflection area is also distorted in the shape reconstructed by NeuS. Collectively, our method outperforms baseline methods by a large margin in all four scenes.

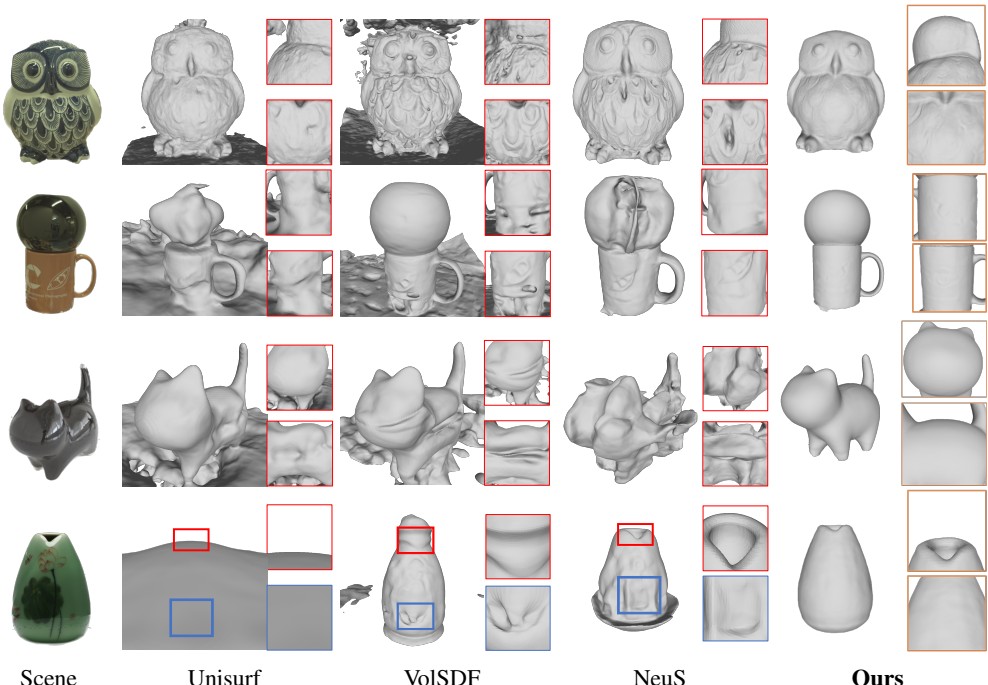

Figure 7: Visual comparison of our method and baseline methods. The Owl and Black Vase scenes are PANDORA Dave et al. (2022) dataset. The Cat scene is sampled with indoor illumination and the Vase with a single light source.

## C.2 COMPARISON ON SYNTHETIC POLREF DATASET

We also evaluated our method on the generated dataset. In contrast to real datasets, ground truth normal maps are accessible in generated scenes, so we introduced the Mean Angular Error (MAE) of normals as an additional evaluation metric. Compared to Chamfer Distance (CD), MAE can reflect the accuracy of reconstructed details better. Results demonstrate that our method still outperforms existing methods for reconstructing reflective objects. Tab. 3 shows our method outperforms both

| Scene | GNeRP (Ours) | | Ref-NeuS | | NeRO | | PANDORA | |
|---|---|---|---|---|---|---|---|---|
| | CD↓ | MAE↓ | CD↓ | MAE↓ | CD↓ | MAE↓ | CD↓ | MAE↓ |
| Bunny | **0.72** | **0.78** | 1.09 | 1.03 | 1.41 | 2.55 | 3.77 | 18.15 |
| Dragon | **0.59** | **1.03** | 0.82 | 1.23 | 2.15 | 3.47 | 5.48 | 10.98 |

Table 3: Comparison on PolReF Synthetic Dataset.

in mesh distance (CD) and accuracy of normals (MAE) quantitatively. Visualization of normals is shown in Fig. 8. Similar to other experiments, the normals reconstructed by existing methods always be over-smoothed, resulting in a lack of details. For instance, the scale details of the dragon scene are omitted in other methods, particularly in the NeRO method. PANDORA fails in both scenes because estimating Stokes vectors with these complex geometries is challenging.

## C.3 VISUAL COMPARISON ON DIFFUSE DATASET

To validate the generalization ability of our method, evaluation of diffuse-dominant objects is conducted on the Camera scene captured in PMVIR (Zhao et al., 2020). Visual comparison with our baseline method NeuS is shown in Fig. 9. Quantitative comparisons are unavailable since the ground truth mesh isn't collected in the dataset.

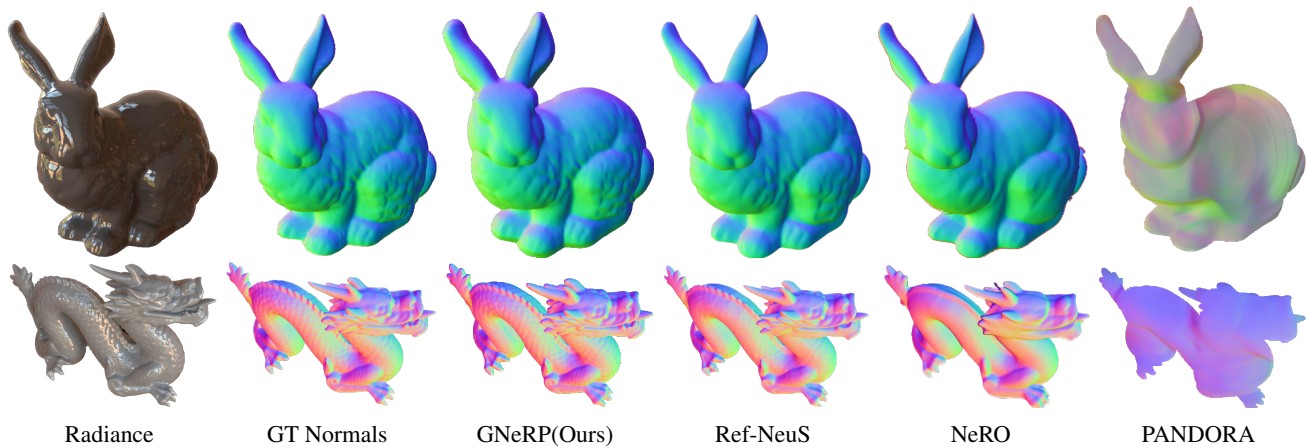

| Radiance | GT Normals | GNeRP(Ours) | Ref-NeuS | NeRO | PANDORA |

Figure 8: Visual comparison of normals on PolRef Synthetic Dataset.

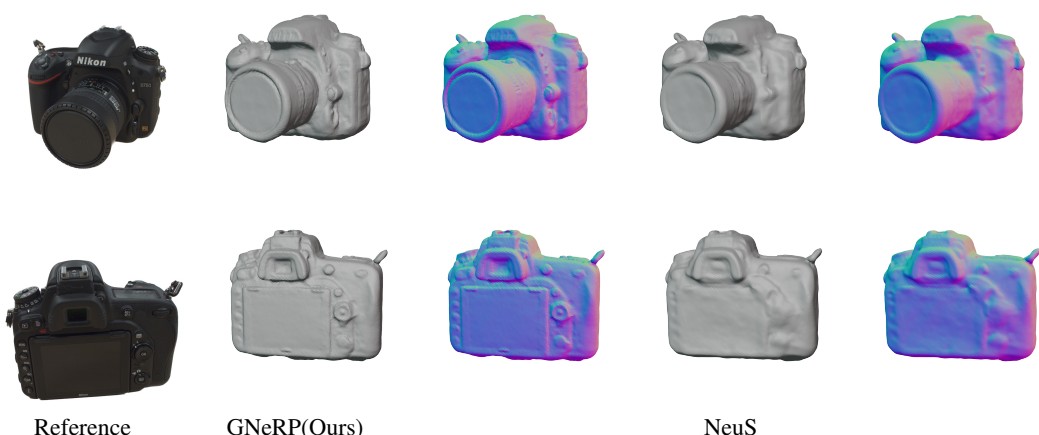

| Reference | GNeRP(Ours) | | NeuS |

Figure 9: Visual comparison of a diffuse object with the baseline method.

As is shown in the figure, our method reconstructs more details of small structures such as buttons, knobs, and slots of the camera. It proves our method can handle diffuse objects concurrently.

### C.4 VISUAL COMPARISON OF NORMALS WITH REF-NERF

Ref-NeRF Verbin et al. (2022) is a state-of-the-art method for reflective object rendering. However, its mesh is inaccessible, and the normals are noisy due to the Integrated Position Encoding (IPE), so we did not involve it in the overall comparison. Instead, we show an example of the Cat scene to show the incomparable normals in Fig.10.

## D ADDITIONAL ABLATION ANALYSIS

### D.1 VISUALIZATION OF EFFECTIVENESS OF DOP REWEIGHTING

DoP efficiently alleviates the noise in polarization prior. One of the most significant scenes is Gnome in the PANDORA dataset, as is shown in Fig. 11.

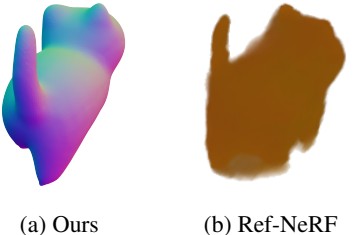

(a) Ours     (b) Ref-NeRF

Figure 10: Normals comparison between Ref-NeRF and ours.

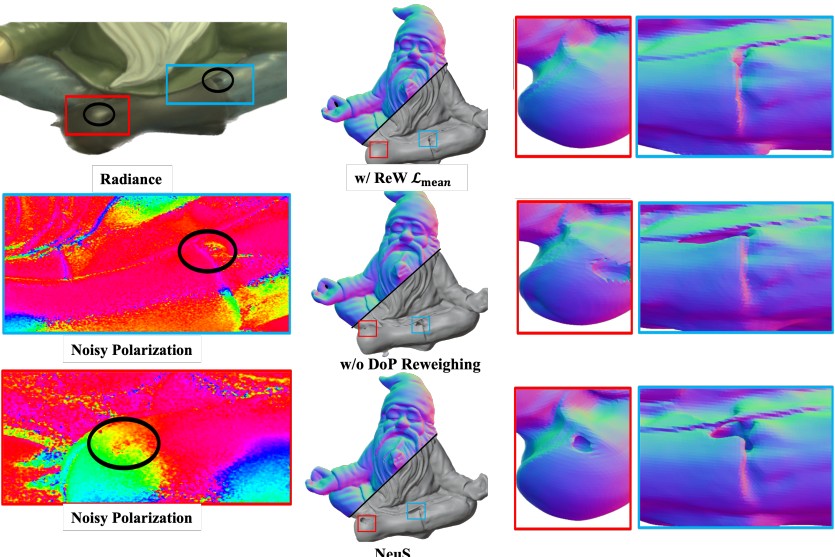

Figure 11: Ablation study of DoP reweighting. Top-Down: w/ DoP ReW., w/o DoP ReW., NeuS

## D.2 VISUALIZATION OF EFFECTIVENESS OF COVARIANCE

$\mathcal{L}_{\text{mean}}$ is our key design regarding to Gaussians of normals. In our experiments, it significantly improves the reconstruction of abrupt normal changes at the mouth of the Duck. It's shown in Fig. 12.

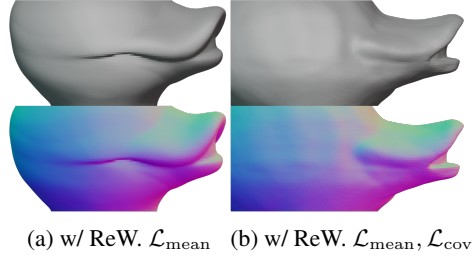

(a) w/ ReW. $\mathcal{L}_{\text{mean}}$    (b) w/ ReW. $\mathcal{L}_{\text{mean}}, \mathcal{L}_{\text{cov}}$

Figure 12: Ablation study on covariance loss.

## D.3 ABLATION STUDY OF HYPER-PARAMETERS

In our experiments, hyperparameters include weights of loss functions ($\alpha$, $\beta$, $\gamma$, and $\delta$) and the number of super-sampled points $M$. $\gamma$ and $\delta$ are fixed at 0.1 to follow previous methods. $\beta$ and $\beta'$

are fixed at $0.1$. However, $\alpha$ is set to either $0.1$ or $1$, depending on the overall ratio of reflection regions. Increased reflection regions indicate a decrease in the reliability of radiance cues, and therefore, the value of $\alpha$ should be decreased.

### D.3.1 WEIGHT OF RADIANCE LOSS

We test different weights of the radiance loss function on the Bunny scene. Results are shown in Fig. 13. Fig. 13 (d) demonstrates that if $\alpha$ is too low, the method cannot extract efficient radiance

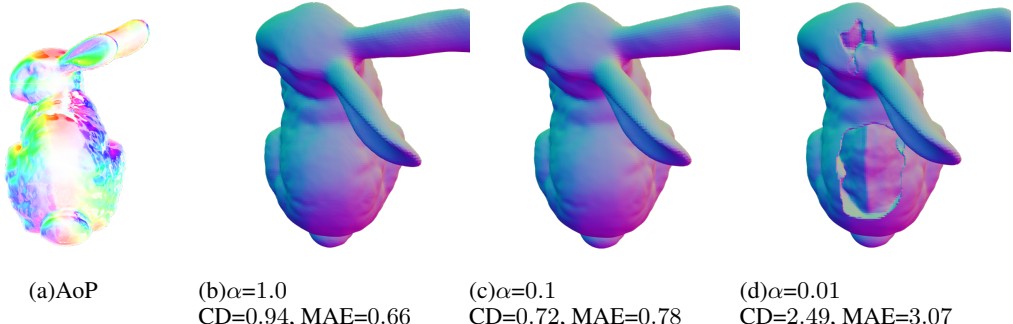

(a)AoP    (b)$\alpha$=1.0 CD=0.94, MAE=0.66    (c)$\alpha$=0.1 CD=0.72, MAE=0.78    (d)$\alpha$=0.01 CD=2.49, MAE=3.07

Figure 13: Ablation study of the weight $\alpha$. CD refers to Chamfer Distance, and MAE refers to Mean Angular Error.

cues, and then the regions with less polarization prior are incorrectly interpreted as holes.

### D.3.2 NUMBER OF SUPER-SAMPLING WITHIN NEIGHBORHOOD

In the paper, $M = 6$ additional points within the neighborhood of $\mathbf{x}_i$ are sampled for simplification. To validate the robustness of this choice, we sampled double points around the ray. Results are shown in Tab. 4. It shows that increasing $M$ will not enhance the accuracy significantly.

| Scene | $M = 6$ | $M = 10$ |
|-------|---------|----------|
| Duck  | 1.63    | 1.62     |

Table 4: Ablation study on $M$.

## E   LIMITATION

We observe that a major limitation of our method is that the reconstruction relies on polarimetric imaging. As illustrated in Fig. 6, radiance images and polarization priors are generated through polarimetric imaging. Moreover, due to hardware limitations, the imaging quality of polarimetric cameras is generally slightly lower compared to regular RGB cameras, and it is more prone to noise in shaded scenes. Furthermore, existing acceleration techniques for NeRF, such as Instant-NGP and Voxel, have not been incorporated into this method. As a result, the training and inference speed of the model is lower compared to existing NGP-based methods.

### E.1 FAILURE CASE

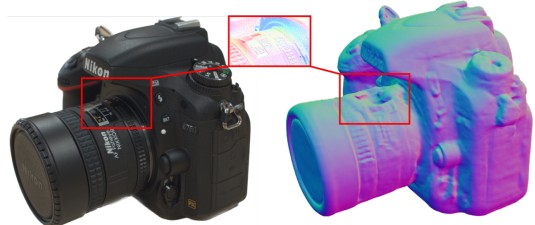

Figure 14: Visualization of distorted mesh resulted by shading.

We present a failure case in Fig. 14. The region bounded by the boxes is shaded by self-occlusion, making the extraction of polarimetric cues noisy shown in the middle sub-figure. Moreover, the radiance is too dim to learn geometry properly. Consequently, a wrong dent exists in the region of the reconstructed mesh.

