# OpenReview forum: "GNeRP: Gaussian-guided Neural Reconstruction of Reflective Objects with Noisy Polarization Priors"
_ICLR.cc/2024/Conference — ICLR 2024 poster_

### Official Review · Reviewer_BtpD · 2023-11-01

**Soundness:** 3 good
**Presentation:** 2 fair
**Contribution:** 3 good
**Rating:** 6
**Confidence:** 3

**Summary:**

This paper proposes a method called GNERP for reconstructing the detailed geometry of reflective objects from multi-view images. The key idea is to extend the geometry representation from scalar Signed Distance Fields (SDFs) to Gaussian fields of normals supervised by polarization priors. The paper introduces a pipeline for learning the surface by volume rendering, and presents a DoP reweighing strategy to alleviate noise and imbalance distribution problems of polarization priors. Additionally, a new multi-view dataset called PolRef, consisting of objects with reflective and less-textured surfaces, is collected to evaluate the performance of3D reconstruction methods. Experimental results show that GNERP improves the geometry details and accuracy of geometry and normals of reflective surfaces compared to existing state-of-the-art methods.

**Strengths:**

- The paper integrates recent advances in Gaussian Splatting to model the polarization priors for reflective surface recovery, which is innovative and aligns with the current research trends.

- The proposed reweighted polarization priors appear to serve effectively as supervision.

**Weaknesses:**

- The explanation of the proposed method is unclear. What is the input? Is a polarization camera required for reconstruction? If not, is the polarization prior regressed during optimization?

- The proposed approach neglects the effect of self-occlusions, which can be severe for complicated objects. The paper should at least include a qualitative discussion about the effects of self-occlusions.

- The property of polarization is closely related to materials. The proposed approach relies on the intermediate output of a coordinate network and hence seems to handle only objects with a uniform material, as shown in Fig. 5.

- The experiments are conducted only on relatively simple objects. It would be beneficial to show results on more complicated objects to better understand its performance.

- It would be great if the authors could provide an illustration of how the polarization priors intuitively improve the performance of NeuS.

**Questions:**

Please see weakness

---

> ### Author Response · Authors · 2023-11-17
>
> Thank you for your comments. We appreciate your acknowledgement of the novelty and effectiveness of our method. We have carefully considered your comments and listed the responses below: (Weakness - W):
>
> W1. We add an illustration to  the Appendix B Figure 6. (a) to show the pipeline of training data preparation.  Please see the training data box for intuition. The inputs include polarization images (AoP and DoP), RGB images, and object masks. A polarization camera is needed for polarization image acquisition, we add the claim to the limitation discussion in Appendix E.
>
> W2. Thanks for pointing out self-occlusion as a potential challenge. We tested additional scenes and found the shading caused by self-occlusion is a limitation of our method, and we added a failure case in the limitation discussion shown in Figure 14.
>
> W3. Thanks for the suggestion of testing scenes with mixed materials. During the discussion time, we processed the raw capture of the Camera scene in PMVIR. It’s a Sony camera with various materials. The reconstruction is shown in Figure 9. As is shown in the figure, our method reconstructs more details of small structures with different materials concurrently. However, the lack of GT mesh in the dataset makes quantitative evaluation unavailable.
>
> W4. Thanks for suggesting testing complicated objects. In the initial submission, the Ironman scene shown in Figure 5 has detailed geometry such as the edges of armours. During the discussion time, we rendered two additional synthetic scenes (Bunny and Dragon) using Mistuba render. The results are shown in Figure 8 and Table 3. As shown in GT normal in Figure 8, The geometry of the Dragon is complicated such as claws, the head with feelers, and scales on the body. It shows our method reconstructs the geometry preserving more details while the geomrty of the other methods are over-smoothed.
>
> W5. Thank you for the suggestion on illustrations. We have illustrated the principle of how Gaussians of normals supervised by polarization improve NeuS in Figure 1. (b). We include the polarization prior in the Figure and modify the caption for clarification.

---

### Official Review · Reviewer_1dU4 · 2023-11-02

**Soundness:** 3 good
**Presentation:** 3 good
**Contribution:** 3 good
**Rating:** 6
**Confidence:** 4

**Summary:**

This paper proposes a neural volume-based method that utilizes Gaussian-based normal representation and polarization information to improve multi-view 3D reconstruction of highly specular objects. Its volume representation and rendering framework are from NeuS. The major novelty is to model the normal direction of sampled 3D points along the ray with 3D Gaussian distributions, which then can be projected to 2D image plane using Gaussian splatting. A new loss is proposed by comparing the projected normal distribution with the normal distribution computed from polarization information. Experiments on datasets containing highly specular objects shows that the proposed method can achieve much more accurate geometry reconstruction.

**Strengths:**

1. High-quality geometry reconstruction for highly specular objects.
Multi-view reconstruction of highly specular surfaces is known to be a difficult problem. This paper shows compelling reconstruction quality on several challenging real-world objects.

2. Convincing improvements compared to prior method on highly specular objects.
Both quantitative and qualitative comparisons show that the proposed method achieve higher geometry reconstruction accuracy compared to prior state-of-the-arts, including a method that also utilizes polarization information and a method designed for specular object.

3. A polarization-based geometry reconstruction method that does not require complicated setups.
The proposed method does not make strong assumptions on the material properties and capturing environment, which makes it practical for many applications.

**Weaknesses:**

1. I have some questions about Sec. 3.2.3, which I would like to understand better. They are not necessarily the weaknesses of the paper.

(a) How to decide the scale factor $s$? Or do we assume the mean loss should be scale invariant and we only care its orientation?

(b) Do we compute $\mu_i^{j}$ by projecting every $x_{i}^{j}$ along the ray to 2D image, i.e. $M'$ equal to $6M$?

(c) I can see that 2D convariance matrices computed from polarization and splatting can both indicate if geometry is changing rapidly in the neighborhood. But it is a bit difficult for me to understand why they should be equal. A side by side visualization of both 2D Gaussian distributions will be useful. In addition, will noisy AoP of diffuse surface cause the 2D covariance matrix from polarization to have very large eigen values?

(d) Typos: there is an extra $)$ in $L_{\text{mean}}$ and $L_{\text{conv}}$.

2. Further ablation studies can be useful to verify the effectiveness of Gaussian normal representation.
Since Gaussian normal representation is the major novelty, it might be good to further verify its effectiveness with more ablation studies. One simple baseline is to render normal through standard volume ray tracing and then use reweighted $L_{\mean}$ loss for training. That may give us a clear answer if we need to sample extra points to compute a Gaussian distribution.

3. Results on less specular objects may be interesting.
I notice that all real objects shown in the paper are quite glossy. Do we have results on less glossy objects? As shown in the paper, the AoP of diffuse surfaces can be noisy but hopefully the DoP weight can fix this issue.

4. Missing reference.
One related paper is cite is: "Sparse Ellipsometry: Portable Acquisition of Polarimetric SVBRDF and Shape with Unstructured Flash Photography, Hwang et al."

Very minor comments:
Figure 2: the polarized direction may use a different color from that of the x-axis. Zenith angle $\theta$ is mentioned by never discussed in the paper. It may be more useful to label out angle $\varphi$.

**Questions:**

My major questions are the first two points listed in the weakness section.
1. Can we explain in more details how to compute 2D normal distribution from a polarized image?
2. Can we further verify if 3D Gaussian distribution is necessary? If we only compute the normal following standard volume ray tracing and then use reweighted $L_{\text{mean}}$ loss, will the results be similar or worse?

**Details Of Ethics Concerns:**

There is no ethic concern as far as I know.

---

> ### Author Response · Authors · 2023-11-17
>
> Thank you for providing feedback. We appreciate your agreement with the performance and weak assumptions of our method. We have carefully considered your comments and listed the responses below (Weakness - W, Question - Q):
>
> W1. (a) We estimated the polarized 2D Gaussians based on unitary projected normals vectors derived from the Angle of Polarization ( the right term of Eq. 8). However, the projected normal vectors are unnecessarily unitary in the left term. The scale factor is multiplied for the equivalence of Eq. 8. The final loss gets rid of the dependent of it, so it isn’t explicitly calculated. We added a detailed description of Gaussian estimation in the Appendix A.3.
>
> (b) In the paper, M and M’ are set to 6 and 4, respectively. To estimate 3D Gaussians, we sample 2 points along the ray (before and after the center) and 4 points along the rays passing adjacent pixels. Hence, when estimating 2D Gaussians in polarization images, we only query the corresponding pixels, and M’ is set to 4.
>
> (c).1 The equivalence is ensured by the relation between the azimuth angle and the Angle of Polarization (AoP), shown in Figure 2 and the equation in the last paragraph of Sec. 3.1 (azimuth angle + pi/2 = AoP mod pi).  So we can recover the orientation of 2D projected normal vectors from AoP, which should align with the orientation of splatted 3D normal vectors. Therefore, covariance should be the same, too. A visualization of 2D Gaussians is Figure 4. (e). Low saturation white pixels indicate that the 2D Gaussian at that location is close to isotropic, resembling a circle. Conversely, high saturation areas indicate strong anisotropy, resembling an ellipse, with the color representing the orientation of the ellipse. This is shown in the left column of the figure.
>
> (c).2 The eigenvalues of covariance will be irregular in diffuse regions due to noisy AoP. Hence we introduce DoP reweighting to reduce their impact since DoP will be small in these regions, as mentioned in Sec. 3.2.3.
>
> (d)Thanks for the error pointed out. We have fixed it.
>
> W2. Thanks for the suggestion on the setting of an additional ablation study. In our understanding, the required experiment is exactly w/ReW. L_{mean} in Table 2.
>
> W3. Thanks for the advice on the evaluation of diffuse scenes. We processed the raw capture of the Camera scene in PMVIR, which is a more diffuse-dominant scene. A visual comparison is shown in Appendix C.3. Since the ground truth mesh isn't provided, quantitative evaluation is unavailable. However, Figure. 9 shows that our method reconstructs more details of small structures such as buttons, knobs, and slots of the camera. This proves that our method can handle diffuse objects effectively.
>
> W4. Thanks for the additional related paper mentioned. We have added it to the Related Work Section.
>
> Minor. We have changed the Figure 2 as the suggestion. The relation between AoP and the normal vector is more clear. Hope it can help you solve W1. (c).
>
> Q1. A detailed explanation of Gaussian calculation is added to the Appendix A.3. 2D Gaussians of polarization images are introduced in the second paragraph.
>
> Q2. We think the question is the same as W2.

---

> > ### Comment · Reviewer_1dU4 · 2023-11-22
> >
> > Thanks for the detailed response! After carefully reading your response, I think all my questions have been solved.

---

### Official Review · Reviewer_Kf6J · 2023-11-06

**Soundness:** 3 good
**Presentation:** 2 fair
**Contribution:** 3 good
**Rating:** 6
**Confidence:** 4

**Summary:**

Neural SDF-based 3D reconstruction excels at smooth Lambertian objects. The paper proposes a 3D Gaussian-based representation of normals in SDF fields, and splat it to 2D Gaussians in the image plane. It shows that the 2D Gaussian can be directly extracted from Angle of Polarization (AoP). This paper proposes to use the proposed 2D Gassians represenation as the additional constraints for 3D normal recovery. Moreover, it proposes to use the Degree of Polarization (DoP) which indicates the complexity of the surface to reweight  AoPalleviate the noise issue of polarization priors.  Their experimental results on PANDORA dataset domstrate the effiectiveness of the proposed method.

**Strengths:**

This paper proposed the 3D Gaussian representation of surface normal for the volume rendering proposed in NeRF(2020) in EQ4., and furture entend it to 2D Gaussian by using the splatting approach (Zwicker et al). The research demonstrates that the 2D Gaussian representation can be efficiently computed using the Angle of Projection (AoP). Additionally, it is revealed that the Degree of Polarization (DoP) is strongly correlated with the surface complexity. Consequently, the authors propose a novel regularization technique for NeRF, involving the reweighting of constraints on the 2D Gaussian representation of surface normals. The experimental results clearly indicate that the proposed method excels in handling high-frequency BRDF surfaces.

**Weaknesses:**

The paper presents experimental results only on the PANDORA dataset. It is essential to include discussions regarding failure cases and the limitations of the proposed method. The authors highlight "the creation of a new and challenging multi-view dataset" as a significant contribution. Nevertheless, there is a notable lack of information and discussion about this new dataset in the paper.

**Questions:**

Please explain in detial about how to get $\hat{\Sigma}, \hat{\Lambda}$ and $\tilde{\Sigma} \tilde{\Lambda}$ in EQ7~9. Pleae explain the experimental setting to get the polarized data. More details are needed for the new multi-view polarized dataset, and how it collected.

---

> ### Author Response · Authors · 2023-11-17
>
> We would like to express our gratitude to the reviewer for his comments and acknowledgments regarding the design of our method and the improved results achieved compared to existing methods. We respond to the weaknesses and questions below.
>
> Weaknesses:
>
> 1. In the initial submission, we presented the results of our PolRef Dataset (Ironman, Snorlax, Cow, Duck, Cat, and Vase) and PANDORA dataset (Owl, Black Vase, and Gnome). During the discussion time, we additionally rendered synthetic scenes of Bunny and Dragon. The supplementary results are now provided in Table 3 and Figure 8. Furthermore, we evaluate our method on a diffuse object (Camera) captured in PMVIR, although ground truth meshes are not available for the PANDORA and PMVIR datasets. Visual comparisons are provided in lieu of quantitative evaluation. They show that our method consistently outperforms existing methods on these datasets.
> 2. We have added the limitations discussion of our method in Appendix E. The primary limitation lies in the requirement of polarimetric imaging, which incurs higher costs compared to RGB cameras. Moreover, the training and inference speeds of our method are lower than those of existing NGP-based reconstruction methods. Moreover, shading caused by self-occlusion may deteriorate the performance. We add a failure case shown in Figure 14.
> 3. We have included a description of our PolRef dataset, along with details of data collection and the evaluation protocol, in Appendix B and Figure 6.
>
> Questions:
>
> 1. We have provided an explanation of the formula to obtain Σ and Λ in Appendix A.3.
> 2. As mentioned in Weakness 3, we have added a description of the dataset collection process in Appendix B.

---

### Official Review · Reviewer_ANa9 · 2023-11-07

**Soundness:** 3 good
**Presentation:** 2 fair
**Contribution:** 3 good
**Rating:** 6
**Confidence:** 4

**Summary:**

The article introduces a 3D reconstruction technique based on Nerf, which focuses on capturing 3D geometries of glossy objects. Building on recent progress in utilizing signed distance functions (SDF), as seen in approaches like NeuS or Ref-Neus, this method also incorporates it. However, unlike the scalar SDF in these approaches, this proposed method additionally integrates polarization cues to constrain the surface normal. These added constraints are formulated using a Gaussian splatting approach. As a result, the proposed method demonstrates notably improved accuracy compared to state-of-the-art (SOTA) approaches.

**Strengths:**

- The primary advantage of the proposed method lies in its utilization of polarization cues to improve the estimated geometry. This strategy is commonly employed to enhance accuracy in conventional 3D reconstruction techniques.
- The additional constraint is devised using a Gaussian splatting technique. Specifically, the original 3D Gaussian on the 3D surface normal is transformed into a 2D Gaussian in the image plane and is constrained by polarization cue.
- The polarization reweighting strategy can guide the proposed method to employ more polarization cues on the area with strong polarization information and employ more radiance on low polarization information.
- The proposed method excels in reconstructing 3D geometry with high precision and capturing intricate details.

**Weaknesses:**

- The proposed method only works for glossy objects that exhibit clear polarization information.
- The proposed method comes with many hyperparameters which are hard for users to set in practice. Parameters like alpha and beta typically vary between 0.1 and 1, contingent on the intricacy of the geometry. Regrettably, the authors did not carry out any experiments to assess the optimal selection of these parameters, potentially hindering the method's practical implementation and performance.
- Only the results of a few objects are shown in the experiments.
- There are typographical errors and improper terms, which are described in the questions.

**Questions:**

- For the parameter selection of alpha and beta, the authors did not specify the use of identical parameters across all experiments. The exact values for these parameters were not explicitly mentioned in the paper.

- In Section 2.4, the authors stated that "the learned 3D Gaussians imply the anisotropic normals distribution of 3D
points and capture more details of surface geometry." Could the authors elaborate on the description or provide some references?
Does it mean the learned 3D Gaussian is anisotropic? And the anisotropicity results in more details?

- The ablation study could include an examination of the impact of alpha and beta. Additionally, the authors did not provide a clear rationale for why they included only two objects in Table 2 while incorporating four objects in Table 1. The ablation study of the other two objects is not supporting?

- In the last paragraph of Section 3.2.3, the authors stated "Intuitively, the first term measures the complexity of the geometry, while the second term reveals the specific geometric shape." Could the authors elaborate on this description?


Additional comments
+ Typographical errors:
	- alpha in the first paragraph of page 5
	- common scenes in the first paragraph of section 2.3
	- Fig. 4(g) in the first paragraph of page 7. In fact, there is no Fig.4 (g).
	- In Fig. 4 caption: "blue boxes bound diffuse ones".


+ "exact 3D shapes" in the abstract should be replaced by another term such as "accurate".

---

> ### Author Response · Authors · 2023-11-17
>
> We thank the reviewer for their insightful comments. We are encouraged that the reviewer acknowledges that our method achieves good results on the typically challenging task of reconstructing reflective objects and outperforms existing methods quantitatively. We respond to the weaknesses and questions below (Weakness 1 - W1, Question 1 - Q1):
>
> W1. To validate the effectiveness of our method on diffuse objects, we processed the raw capture of the Camera scene in PMVIR. A visual comparison is shown in Appendix C.3. Since the ground truth mesh isn't provided, quantitative evaluation is unavailable. However, Fig. 9 shows that our method reconstructs more details of small structures such as buttons, knobs, and slots of the camera. This proves that our method can handle diffuse objects effectively.
>
> W2. The setting of hyper-parameters is relatively fixed in our method. For clarification, we rearrange the notation of the weights of loss functions in Eq. 9 and add a description of the settings in Appendix D.3. The only hyper-parameter that needs to be tuned is alpha. We tested different choices of alpha on the Bunny scene, as shown in Appendix D.3.2, to facilitate the practical implementation of the method.
>
> W3. In the initial submission, we showed reconstructed meshes of the Owl, Black Vase, Cat, and Vase scenes in the Appendix. Moreover, we rendered two additional synthetic scenes (Bunny and Dragon)  with ground truth normals during the discussion time, and qualitative and quantitative comparisons are shown in Appendix C.2, along with an additional evaluation metric, Mean Angular Error (MAE) of normals.
>
> W4. Thank you for pointing out the errors. We have fixed them.
>
> Q1. As mentioned in W2, we list them in Appendix D.3.
>
> Q2. Yes, as parameterized by the covariance matrix, the learned Gaussians are anisotropic. Normal vector can be seen as the mean of 3D Gaussians, and changes of normal vectors within the neighborhood are captured by covariance of Gaussians. Therefore, we claim that the 3D Gaussians capture more details. We have modified the description in Sec. 2.4.
>
> Q3. As mentioned in W2, the required experiments are listed in Appendix D.3.2. Due to the limitation of computational resources, ablation studies were only done on two of the objects in Table 1. We plan to complete them when the computational resources become available. We’ll post them if the results are available during the discussion.
>
> Q4. The first term supervises the eigenvalues of the covariance matrix, and the second term supervises the eigenvectors of the covariance matrix. Similar to PCA techniques, the covariance matrix determines the magnitude and direction of normal changes through eigenvalues and eigenvectors. If the local shape is like a plane, normals will change smoothly in all directions, and the difference between eigenvalues will be small, resulting in the Anisotropy (defined as the ratio of eigenvalues in the paper) approaching 1. If there are some details like edges, normals tend to change abruptly and exhibit directionality, which is represented by eigenvectors. We have modified the description in Sec. 3.2.3.

---

### Meta-Review · Area_Chair_emU5 · 2023-12-05

**Metareview:**

This paper describes a 3D reconstruction method based on a neural volume-based method. The proposed method extends NeuS by utilizing Gaussian-based normal representation to improve multi-view 3D reconstruction of highly specular surfaces. The key idea is on the new representation; it extends the geometry representation from scalar SDFs to Gaussian fields of normals supervised by polarization priors. The result demonstrates the effectiveness of the proposed method.

The major strengths of the paper are:
1) The new geometry representation that is effective for reconstructing shiny surfaces.
2) It showed a strong correlation between the polarization prior and surface complexity.
3) Compelling results are obtained by the proposed method.

The weakness, on the other hand, is that the effectiveness for diffuse surfaces/mixed materials is not fully showcased. There is still a remaining question about it.

Overall, the reviewers are positive about the paper because of the nice integration of NeuS + polarization prior. The merit surpasses the negatives. The reviewers and AC read the rebuttal and took it into consideration for the final recommendation.

**Justification For Why Not Higher Score:**

Although the method's novelty is appreciated, there remained a question about whether the method works well for more diverse materials. If the paper covered this aspect better, the scores could be higher.

**Justification For Why Not Lower Score:**

The novelty of the idea is well appreciated, and reviewers and AC agree that the idea of extending the scalar SDFs to vector-valued fields is somehow interesting and worth sharing.

---

### Decision · Program_Chairs · 2024-01-16

Accept (poster)